

# Field theories for type-II fractons

**Weslei B. Fontana**[1,2⋆], **Pedro R. S. Gomes**[2†] **and Claudio Chamon**[1‡]

**1** Physics Department, Boston University, Boston, MA, 02215, USA
**2** Departamento de Física, Universidade Estadual de Londrina,
86057-970, Londrina, PR, Brasil

⋆ weslei@uel.br, † pedrogomes@uel.br, ‡ chamon@bu.edu

## Abstract

We derive an effective field theory for a type-II fracton starting from the Haah code on the lattice. The effective topological theory is not given exclusively in terms of an action; it must be supplemented with a condition that selects physical states. Without the constraint, the action only describes a type-I fracton. The constraint emerges from a condition that cube operators multiply to the identity, and it cannot be consistently implemented in the continuum theory at the operator level, but only in a weaker form, in terms of matrix elements of physical states. Informed by these studies and starting from the opposite end, i.e., the continuum, we discuss a Chern-Simons-like theory that does not need a constraint or projector, and yet has no mobile excitations. Whether this continuum theory admits a lattice counterpart remains unanswered.



# 1  Introduction

Fracton topological order, originally constructed in lattice spin models [1–16] and later on extended to the scope of continuum field theories [17–31], is characterized by a gapped spectrum with quasiparticle excitations with either restricted mobility (type-I fractons) or no mobility at all (type-II fractons), and a ground state degeneracy (GSD) that depends not only on the topology of the manifold but also on the geometry of the lattice. This dependence on the lattice details signals a sort of ultraviolet/infrared (UV/IR) mixing, i.e., fracton systems do not present the usual decoupling between high and low-energy physical properties. These exotic properties of fractonic systems make the problem of finding effective low-energy theories rather interesting. Recently, continuum field theories that capture these unusual properties have been successfully constructed for gapped systems with fracton excitations of type-I. Field theory descriptions of gapless systems with fracton excitations of type-I and II have also been obtained. However, a field theory description of fracton topological order of type-II – a gapped theory with completely immobile excitations – is thus far missing.

The purpose of this paper is to construct effective field theories for gapped type-II fractons. We follow a UV-to-IR prescription that starts from a lattice model for a type-II fracton, specifically the Haah code [3]. Our construction benefits from insights from previous studies of effective field theories inspired by the Haah code [32–34]. These constructions capture some of its physical properties, but the resulting effective theories still contain mobile excitations and are gapless. As we shall discuss, to ensure that all excitations are completely immobile, it is essential that the theory contains infinitely many charge conservation laws that prevent excitations from moving. Equivalently, the immobility is connected to the impossibility of constructing gauge invariant space-like line operators that represent trajectories of excitations. The gaplessness of previous constructions comes from the Maxwell-like form of the field theory; the effective theory we deduce here is instead of a Chern-Simons form and hence gapped.

One of the main difficulties in obtaining an effective field theory for type-II fracton topological order is that, in trying to construct a fully gapped gauge theory, we immediately run into a problem. In a fully gapped gauge theory, we can in principle construct line operators which in turn describe trajectories of excitations. However, this is in contradiction with the most salient feature of a type-II theory: that all the excitations are completely immobile. In our approach we overcome this difficult. Starting from the lattice, we represent the microscopic operators in terms of fields that are well-defined in the continuum limit and that lead to a fully gapped gauge theory. However, the type-II effective field theory is not given exclusively in terms of the action; it must be supplemented with a condition that selects physical states. This condition emerges from the requirement that the continuum operators properly reproduce the features of the lattice ones. More precisely, this condition implements the property that the product of all spin operators contained in a cube operator (see Fig. 1 below) reduces to the identity. As we shall discuss, this feature of the lattice cannot be consistently implemented in the continuum theory at the operator level, but only in a weaker form, in terms of matrix elements of physical states, i.e., as a criterion for selecting physics states. By itself, the action we have obtained describes a type-I fracton topological order, since the associated Hilbert space contains mobile excitations. It is only inside the physical subspace that the type-II fracton properties manifest. In this sense, the type-II fracton character is embedded into a type-I theory.



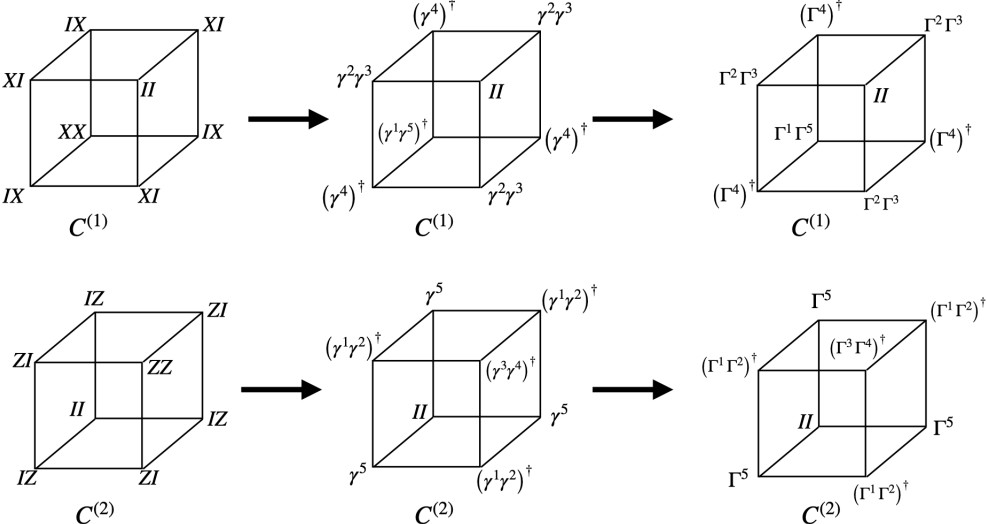

Figure 1: Cube operators of the Haah code. The notation indicates the product of Pauli operators in each site, e.g., $ZZ \equiv \sigma^Z \otimes \sigma^Z$. The $\gamma^I$ are the associated Dirac operators. The $\Gamma^{(I,\alpha)}$ of each cube are equivalent to $\gamma^I$ but needed for a consistent effective theory. The $\alpha$ index is supressed in the figure for simplicity.

## 2  Haah Code and Its Effective Field Theory

The Haah model is defined on a the 3-dimensional cubic lattice $\Lambda$ [3]. Its Hamiltonian is written in terms of two types of cube operators, $C_{\mathbf{x}}^{(1)}$ and $C_{\mathbf{x}}^{(2)}$:

$$H_{\text{Haah}} = -\sum_{\mathbf{x} \in \Lambda^*} C_{\mathbf{x}}^{(1)} - \sum_{\mathbf{x} \in \Lambda^*} C_{\mathbf{x}}^{(2)}, \tag{1}$$

where $\mathbf{x}$ labels the cubes (or sites in the dual cubic lattice $\Lambda^*$). The cube operators are constructed as products of two Pauli operators at each of the eight corners of the cube, as depicted in Fig. 1.

One can construct the corresponding low-energy effective theory by following the procedure presented in [35]. To this end, we write the cube operators in the Dirac basis. Consider the set of anti-commuting operators $\gamma^I$, $I \in S^{(2)}$, with $S^{(2)} = \{12, 22, 32, 01, 03\}$, where the notation $ij$ represents the tensor product $\sigma_i \otimes \sigma_j$, with $\sigma_0 \equiv \mathbb{1}$. In terms of this basis of Dirac $\gamma^I$ matrices, the cube operators become

$$C_{\mathbf{x}}^{(1)} = \left(\gamma^1 \gamma^5\right)^{\dagger}_{\mathbf{x}-\hat{x}-\hat{y}-\hat{z}} \left(\gamma^4\right)^{\dagger}_{\mathbf{x}-\hat{x}+\hat{y}-\hat{z}} \left(\gamma^4\right)^{\dagger}_{\mathbf{x}+\hat{x}-\hat{y}-\hat{z}} \left(\gamma^4\right)^{\dagger}_{\mathbf{x}-\hat{x}-\hat{y}+\hat{z}}$$
$$\times \left(\gamma^2 \gamma^3\right)_{\mathbf{x}+\hat{x}+\hat{y}-\hat{z}} \left(\gamma^2 \gamma^3\right)_{\mathbf{x}-\hat{x}+\hat{y}+\hat{z}} \left(\gamma^2 \gamma^3\right)_{\mathbf{x}+\hat{x}-\hat{y}+\hat{z}} \mathbb{1}_{\mathbf{x}+\hat{x}+\hat{y}+\hat{z}}, \tag{2}$$

and

$$C_{\mathbf{x}}^{(2)} = \mathbb{1}_{\mathbf{x}-\hat{x}-\hat{y}-\hat{z}} \, \gamma^5_{\mathbf{x}-\hat{x}+\hat{y}-\hat{z}} \, \gamma^5_{\mathbf{x}+\hat{x}-\hat{y}-\hat{z}} \, \gamma^5_{\mathbf{x}-\hat{x}-\hat{y}+\hat{z}}$$
$$\times \left(\gamma^1 \gamma^2\right)^{\dagger}_{\mathbf{x}+\hat{x}+\hat{y}-\hat{z}} \left(\gamma^1 \gamma^2\right)^{\dagger}_{\mathbf{x}-\hat{x}+\hat{y}+\hat{z}} \left(\gamma^1 \gamma^2\right)^{\dagger}_{\mathbf{x}+\hat{x}-\hat{y}+\hat{z}} \left(\gamma^3 \gamma^4\right)^{\dagger}_{\mathbf{x}+\hat{x}+\hat{y}+\hat{z}}. \tag{3}$$

Our choice of conjugation of certain operators in both cubes is completely innocuous since all the lattice operators are Hermitian. This is simply a convenience for the description in terms of continuum fields, leading to more symmetrical forms for the $T$-vectors that will be constructed below. Which operators are conjugated or not reflects assignments of charges at the corners of the cube. According to the choices in Fig.1, under an inversion about the center, the charge

arrangement of the cube $C^{(1)}$ is mapped into the one of the cube $C^{(2)}$ and vice-versa. This symmetry mimics the self-duality between the cube operators.

We then follow the procedure described in [35] and represent the Dirac matrices as

$$\gamma_{\mathbf{x}}^{(I)} \equiv \exp\left(i\, t_a^{(I)} K_{ab} A_b(\mathbf{x})\right), \tag{4}$$

where $a, b = 1, ..., 4$ (We need four independent fields $A_a$ to realize a four dimensional representation of the Clifford algebra). Commutation relations between the fields $A_a(\mathbf{x})$ are set in order to reproduce the algebra of Dirac matrices,

$$[A_a(\mathbf{x}), A_b(\mathbf{x}')] \equiv i\pi (K^{-1})_{ab}\, \delta_{\mathbf{x},\mathbf{x}'}. \tag{5}$$

This, in turn, implies that the matrix $K$ and the set of vectors $t_a^{(I)}$ must satisfy

$$t_a^{(I)} (K^\top)_{ab}\, t_b^{(J)} = \begin{cases} 1 \mod 2, & I \neq J, \\ 0 \mod 2, & I = J, \end{cases} \tag{6}$$

to ensure that the representation in (4) reproduces properly the anticommutation relations of the $\gamma_{\mathbf{x}}^{(I)}$ matrices. For an antisymmetric matrix $K$, the condition in second line is satisfied exactly (not mod 2). A simple choice for $K$ and $t_a^{(I)}$ is $K_{ab} \equiv +1$ if $a < b$ and the vectors $t_a^{(I)}$, $I = 1, ..., 5$ defined as $t_a^{(I)} \equiv \delta_a^I$ for $I = 1, ..., 4$ and $t_a^{(5)}$ defined so that the neutrality condition $\sum_{I=1}^5 t_a^{(I)} = 0$ is satisfied (this is equivalent to $\gamma^1 \gamma^2 \gamma^3 \gamma^4 \gamma^5 = \mathbb{1}$). These vectors identify the elements of $S^{(2)}$, and we refer to them as the *principal basis*.

The conditions (6) imply that the fields $A_a(\mathbf{x})$ in (4) are $U(1)$ compact, since the exponential is unchanged under the shifts

$$A_a \to A_a + 2\pi \sum_{J=1}^3 t_a^{(J)} m_J, \quad m_J \in \mathbb{Z}. \tag{7}$$

Under these shifts the exponential changes by the factor

$$\exp\left(2\pi i \sum_{J=1}^3 t_a^{(I)} K_{ab}\, t_b^{(J)} m_J\right), \tag{8}$$

which is equal to one due to (6).

To each corner of the cubes we assign a $\Gamma^{(I,\alpha)}$ operator

$$\Gamma^{(I,\alpha)} \equiv \left(\gamma^1\right)^{T_1^{(I,\alpha)}} \left(\gamma^2\right)^{T_2^{(I,\alpha)}} \left(\gamma^3\right)^{T_3^{(I,\alpha)}} \left(\gamma^4\right)^{T_4^{(I,\alpha)}}, \tag{9}$$

where the index $\alpha = 1, 2$ indicates whether the operator belongs to either $C^{(1)}$ or $C^{(2)}$, and $T^{(I,\alpha)}$ can be written as a linear combination of the principal basis vectors. They have integer entries, which are defined only mod 2. (The freedom mod 2 arises from the fact that $(\gamma^I)^2 = \mathbb{1}$.) We stress that the identification between $T^{(I,\alpha)}$ and $\Gamma^{(I,\alpha)}$ is unique on each cube.

The microscopic description has further properties that will be essential to arrive at a consistent low-energy effective field theory. These properties constrain the allowed $T$-vectors in the continuum theory [1]. They are summarized as follows:

---

[1] While the condition that the vectors $t^{(I)}$ are defined mod 2 are sufficient to ensure the proper commutation relations between Dirac matrices, the gauge invariance in the continuum requires that conditions like $\mathcal{T}_a^{(I,\alpha)} K_{ab} \mathcal{T}_b^{(J,\beta)} = 0$ are satisfied exactly, where $\mathcal{T}_a^{(I,\alpha)}$ are linear combinations of $T^{(I,\alpha)}$. These relations cannot be satisfied by the principal basis vectors $t_a^{(I)}$, but can be adjusted with $t^{(I)}$ mod 2. This is the reason we are forced to introduce the operators (9).

(i) **The $\Gamma^{(I,\alpha)}$ operators are equivalent to the $\gamma^I$, i.e.,**

$$T_a^{(I,\alpha)} = t_a^{(I)} \mod 2 \quad \text{and} \quad \sum_{I=1}^{5} T_a^{(I,\alpha)} = 0. \tag{10}$$

This condition allows for the entries of $T^{(I,\alpha)}$ to differ from the entries of $t^{(I)}$ up to the addition of an even integer, but the sum of them satisfies the neutrality condition exactly (not only mod 2), as the principal basis does.

(ii) **The cube operators commute, $[C_{\mathbf{x}}^{(\alpha)}, C_{\mathbf{x'}}^{(\beta)}] = 0$, for all $\alpha, \beta = 1, 2$, and x, x$'$.** The commutation constrains the allowed $T$-vector to obey a set of relations. For example, the commutation relation $[C_{\mathbf{x}}^{(1)}, C_{\mathbf{x}-\hat{x}+\hat{y}+\hat{z}}^{(1)}] = 0$ implies that $\left(T^{(2,1)} + T^{(3,1)}\right) K\, T^{(4,1)} = 0$ mod 2. There are in total 12 such relations that can be read systematically from Fig.(1) and are listed explicitly in Appendix A.

(iii) **The eight operators on the corners of each cube multiply to the identity.** This constraint is equivalent to requiring that the $T$-vectors at the corners of the cubes in Fig.(1) add to zero mod 2:

$$-T_a^{(1,1)} - T_a^{(5,1)} + 3\left(T_a^{(2,1)} + T_a^{(3,1)} - T_a^{(4,1)}\right) = 0 \quad \mod 2,$$
$$-T_a^{(3,2)} - T_a^{(4,2)} + 3\left(T_a^{(5,2)} - T_a^{(1,2)} - T_a^{(2,2)}\right) = 0 \quad \mod 2. \tag{11}$$

Demanding that the conditions (ii) resulting from the commutation of the cubes vanish exactly (not only mod 2) ensures gauge invariance of the low-energy effective theory, similarly to the case of the type-I fractons studied in [35]. The following choices satisfies the conditions (i) and (ii):

$$T_a^{(1,1)} = \begin{pmatrix} 1 \\ 0 \\ 0 \\ 0 \end{pmatrix}, \quad T_a^{(2,1)} = \begin{pmatrix} 0 \\ 1 \\ 0 \\ 0 \end{pmatrix}, \quad T_a^{(3,1)} = \begin{pmatrix} 0 \\ 0 \\ -1 \\ 0 \end{pmatrix}, \quad T_a^{(4,1)} = \begin{pmatrix} 0 \\ 0 \\ 0 \\ 1 \end{pmatrix}, \quad T_a^{(5,1)} = \begin{pmatrix} -1 \\ -1 \\ 1 \\ -1 \end{pmatrix}, \tag{12}$$

and

$$T_a^{(1,2)} = \begin{pmatrix} 1 \\ 0 \\ 0 \\ 0 \end{pmatrix}, \quad T_a^{(2,2)} = \begin{pmatrix} 0 \\ -1 \\ 0 \\ 0 \end{pmatrix}, \quad T_a^{(3,2)} = \begin{pmatrix} 0 \\ 0 \\ 1 \\ 0 \end{pmatrix}, \quad T_a^{(4,2)} = \begin{pmatrix} 0 \\ 0 \\ 0 \\ 1 \end{pmatrix}, \quad T_a^{(5,2)} = \begin{pmatrix} -1 \\ 1 \\ -1 \\ -1 \end{pmatrix}. \tag{13}$$

However, these vectors do not satisfy condition (iii) exactly, but only mod 2. Therefore, we cannot implement (iii) at the operator level[2], instead we impose the conditions in (11) as constraints in the Hilbert space of the continuum theory. The specific form of the $T$-vectors is not unique. Other choices for these vectors are allowed as long they are compatible with conditions (i) and (ii). The constraint imposed by condition (iii) would also change accordingly, but there would not be any physical consequence in the continuum description. In this sense, there exists an equivalence class of $T$-vectors that defines a consistent effective theory.

The exponential map (4) allows us to write the cube operators in the continuum limit,

$$C_{\mathbf{x}}^{(\alpha)} \equiv \exp\left[i\,\mathcal{T}_a^{(1,\alpha)} K_{ab} A_b\right] \exp\left[i\left(\mathcal{T}_a^{(1,\alpha)} K_{ab}\, d_1 A_b + \mathcal{T}_a^{(2,\alpha)} K_{ab}\, d_2 A_b\right)\right] + h.c, \tag{14}$$

---

[2]In fact, conditions (i), (ii) and (iii) are mutually exclusive and cannot be simultaneously satisfied. Any set of $T$-vectors obeying two of the three conditions automatically breaks the third one.

where the derivatives $d_i$ are defined as

$$\begin{pmatrix} d_1 \\ d_2 \end{pmatrix} \equiv \begin{pmatrix} \frac{1}{2}\left(\partial_x^2 + \partial_y^2 + \partial_z^2\right) \\ \partial_x\,\partial_y + \partial_y\,\partial_z + \partial_x\,\partial_z \end{pmatrix}. \tag{15}$$

The charge vectors $\mathcal{T}_a^{(i,\alpha)}$ are expressed as a combination of the $T_a^{(I,\alpha)}$. They are given explicitly by

$$\mathcal{T}^{(1,1)} = \begin{pmatrix} 0 \\ 4 \\ -4 \\ -2 \end{pmatrix}, \quad \mathcal{T}^{(2,1)} = \begin{pmatrix} 0 \\ 0 \\ 0 \\ 2 \end{pmatrix} \quad \text{and} \quad \mathcal{T}^{(1,2)} = \begin{pmatrix} -6 \\ 6 \\ -4 \\ -4 \end{pmatrix}, \quad \mathcal{T}^{(2,2)} = \begin{pmatrix} 2 \\ -2 \\ 0 \\ 0 \end{pmatrix}. \tag{16}$$

We remark that $\mathcal{T}^{(1,\alpha)} K \mathcal{T}^{(2,\alpha)} = 0$ and hence the terms in the exponentials in (14) commute.

The term in the first exponential in (14) originates from the product of the eight operators in the corners of the cubes in condition (iii) above. As the conditions in (11) cannot be enforced to vanish exactly in the continuum theory, we impose them in terms of matrix elements. In other words, we select the physical states via the conditions

$$\mathcal{T}_a^{(1,\alpha)} K_{ab} A_b \,|\text{phys}\rangle = 0, \quad \alpha = 1, 2. \tag{17}$$

We shall consider the deformed (enlarged) theory obtained by omitting the first exponential in (14), and recover the physical subspace by means of (17). At the first sight, the selection rule (17) does not define consistently a subspace which is closed under time evolution, since it does not commute with the full Hamiltonian. However, we will use it only after the theory is properly projected onto the ground state (in the low-energy effective theory), where the closure under time evolution is trivial.

We shall assign a charge $q^{(\alpha)}$ for each cube excitation (corresponding to the eigenvalue $c_{\vec{x}}^{(\alpha)} = -1$), i.e, $c_{\vec{x}}^{(\alpha)} = -1 \Leftrightarrow q^{(\alpha)}$. As the eigenvalues of the cube operators are $\pm 1$, then the charges in the continuum are defined $q^{(\alpha)} \bmod 2q^{(\alpha)}$. Note that since $q^{(\alpha)}$ is not necessarily unit, we need to impose that the charge is defined $\bmod\, 2q^{(\alpha)}$ to reproduce the $\mathbb{Z}_2$ charge of the lattice model. We will discuss the excitations of the continuum action in more detail in section(3.2) .

The redefined cube operators can be written as

$$C_{\mathbf{x}}^{(\alpha)} = \exp\left[i K_{ab}\,\mathcal{D}_a^{(\alpha)} A_b\right] + h.c\,, \tag{18}$$

where

$$\mathcal{D}_a^{(\alpha)} \equiv \sum_{i=1}^2 \mathcal{T}_a^{(i,\alpha)} d_i\,. \tag{19}$$

In the continuum limit the Hamiltonian of the enlarged theory can be written as

$$H \sim -\sum_\alpha \int d^3x\,\cos\left(\mathcal{D}_a^{(\alpha)} K_{ab} A_b\right). \tag{20}$$

The ground state corresponds to the situation where all the cosines are maximized. This can be enforced through a Lagrange multiplier $A_0^{(\alpha)}$ for each of the cubes. We thus arrive at the enlarged low-energy effective theory

$$S = \int d^3x\,dt\left[\frac{1}{2\pi}A_a K_{ab}\,\partial_0 A_b + \frac{1}{\pi}A_0^{(\alpha)} K_{ab}\,\mathcal{D}_a^{(\alpha)} A_b\right]. \tag{21}$$

The first term of the action gives the equal-time commutation relation (5) while the second one corresponds to the ground state constraint. We recall that the low-energy physical subspace is obtained upon using (17). The action (21) is invariant under the gauge transformations

$$A_0^{(\alpha)} \to A_0^{(\alpha)} + \partial_0 \zeta^{(\alpha)}, \tag{22}$$

$$A_a \to A_a + \mathcal{D}_a^{(\alpha)} \zeta^{(\alpha)}, \tag{23}$$

provided that $K_{ab} \mathcal{D}_a^{(\alpha)} \mathcal{D}_b^{(\beta)} = 0$, which follows directly from the requirements in (ii) above. In the following we shall discuss some properties of the effective field theory.

# 3 Aspects of the effective field theory

## 3.1 Level Quantization

The matrix $K$ entering the low-energy effective action (21) plays the role of a level, just as in the case of usual Chern-Simons theories. A natural question is whether it carries some notion of quantization. Coming from the lattice, the elements of the matrix $K$ were chosen to be quantized. This was just a simple solution of the conditions in (6). Of course, the choice of $t_a^{(I)}$ and $K$ is not unique. Therefore, we can think of the conditions in (6) as providing certain quantization conditions for the elements of the matrix $K$, given a set of vectors $t_a^{(I)}$. For example, with the vectors $t_a^{(I)}$ in the principal basis, the conditions (6) translate into the following quantization condition for the elements of the matrix $K$:

$$K_{IJ} = \text{odd}, \quad I \neq J \quad \text{and} \quad I, J = 1, \dots, 4. \tag{24}$$

The next natural question is whether one could extract some notion of quantization exclusively from the effective theory (21) without making reference to the lattice, relying only on the possible existence of certain types of large gauge transformations. While a positive answer might be expected in view of the fact that the level is quantized in our case (from the lattice), and also from the analogy with usual Chern-Simons theory, we could not find a way to show level quantization directly from the continuum theory.

## 3.2 Excitations and Their Immobility in the Effective Field Theory

We can use the effective action (21) to study the low-energy properties of the Haah code after proper projection onto the physical subspace. We consider low-lying excitations parametrized by currents and couple them to the gauge fields according to $\int d^3x \, dt [J_0^{(\alpha)} A_0^{(\alpha)} + J_a A_a]$, which is gauge invariant provided that the current is conserved

$$\partial_0 J_0^{(\alpha)} = \mathcal{D}_a^{(\alpha)} J_a. \tag{25}$$

Note that the spatial currents $J_a$ have no flavor index, we interpret that as meaning that the excitations in each cube are not completely independent from each other. This mimics a feature of the lattice model. For example, action of a $IY$ operator on a given site produces a tetrahedral configuration on each cube (see Fig.1 for reference). This is incorporated in the effective theory through (25) with $J_a$ carrying no flavor index. This continuity relation also leads to infinitely many conserved charges [32]

$$Q_f^{(\alpha)} = \int d^3x \, f(\mathbf{x}) J_0^{(\alpha)}, \tag{26}$$

provided that $\mathcal{D}_a^{(\alpha)} f(\mathbf{x}) = 0$. Note that the trivial case $f = $ constant gives the common notion of a global charge. Below we show that the projection onto the physical Hilbert space ensures the immobility of all excitations (in Appendix B, we show that in the unconstrained Hilbert space there is dipole mobility along the (111) direction). To properly project onto the physical subspace consider the equation of motion of $A_0^{(\alpha)}$, which provides a "flux-attachment" relation:

$$J_0^{(\alpha)} = \frac{1}{\pi} K_{ab} \, \mathcal{D}_a^{(\alpha)} A_b \,. \tag{27}$$

We can use this relation to examine the corresponding conservation laws inside the physical subspace. Denoting the physical states generically as $|\text{phys}\rangle$, the relation (27) leads to

$$
\begin{aligned}
\frac{d}{dt} Q_f^{(\alpha)} |\text{phys}\rangle &= \int d^3x \, f(\mathbf{x}) \frac{1}{\pi} K_{ab} \left[ \mathcal{T}_a^{(1,\alpha)} d_1 + \mathcal{T}_a^{(2,\alpha)} d_2 \right] \partial_0 A_b \, |\text{phys}\rangle \\
&= \int d^3x \, f(\mathbf{x}) \frac{1}{\pi} K_{ab} \, \mathcal{T}_a^{(2,\alpha)} d_2 \, \partial_0 A_b \, |\text{phys}\rangle \,,
\end{aligned}
\tag{28}
$$

where we have used the constraint $\mathcal{T}_a^{(1,\alpha)} K_{ab} A_b |\text{phys}\rangle = 0$. Therefore, inside the physical subspace, the function $f(\mathbf{x})$ is less restricted than in the full Hilbert space, since it only needs to satisfy $d_2 f = 0$, instead of $d_1 f = d_2 f = 0$. This, in turn, implies a more general set of conserved charges and consequently more restrictions on the mobility of the excitations, leading ultimately to the complete immobility of all quasiparticles.

To highlight the difference between the constrained and unconstrained cases, let us first consider solutions of $d_1 f = d_2 f = 0$, which can be generically written as $f = (c_1 l + c_0) h(u, v)$, where $h(u, v)$ is a harmonic function and the coordinates $(l, u, v)$ are explicitly given by

$$\hat{l} \equiv \frac{1}{\sqrt{3}} (\hat{x} + \hat{y} + \hat{z}) \,, \quad \hat{u} \equiv \frac{1}{\sqrt{2}} (\hat{y} - \hat{z}) \,, \quad \hat{v} \equiv \frac{1}{\sqrt{6}} (-2\hat{x} + \hat{y} + \hat{z}) \,. \tag{29}$$

For example, the density $J_0^{(\alpha)} = q^{(\alpha)} \delta(u) \delta(v) [\delta(l - l_0(t)) - \delta(l - l_1(t))]$ is such that for any function $f$ of the form above, $Q_f^{(\alpha)} = c_1 (l_0 - l_1) q^{(\alpha)} h(0, 0)$ is conserved provided that $l_0 - l_1$ is constant. The density $J_0^{(\alpha)}$ corresponds to a dipole moving along the (111) direction. Therefore, we find that the restrictions $d_1 f = d_2 f = 0$ allow for mobile excitations.

In contrast, when the condition $d_1 f = 0$ is no longer required, the space of solutions for $f$ is much less restricted (e.g., not forced to have at most linear dependence on $l$). For example, $f = c_1 x$ implies dipole conservation along the (100) direction, and similarly for dipole conservation along (010) and (001) directions with $f = c_2 y$ and $f = c_3 z$. The function $f = c_{11} x^2$ implies conservation of one of the components of the quadrupole tensor. In appendix (C) we construct multinomial solutions of arbitrary order. These infinitely many conservation laws is what prevents mobility of excitations.

The mobility (or lack of thereof) can be understood in an equivalent way by means of gauge invariant line operators. In the full Hilbert space with both $d_1$ and $d_2$ operators, it is always possible to build a line operator $W = \exp\left(i \int_{\mathcal{C}} \tilde{A}_a\right)$, using a linear combination $\tilde{A}_a$ of the gauge fields, where $\mathcal{C}$ is a path along the (111) direction. For example, the linear combinations $\tilde{A}_1 = -\frac{1}{3}(A_1 + \frac{2}{3} A_2 + \frac{2}{3} A_3)$ and $\tilde{A}_2 = \frac{2}{3}(A_2 + \frac{1}{2} A_3 + \frac{1}{2} A_4)$ transform as

$$\delta \tilde{A}_1 = \partial_l^2 \zeta^{(2)} \quad \text{and} \quad \delta \tilde{A}_2 = \partial_l^2 \zeta^{(1)} \,, \tag{30}$$

so that the corresponding line operators are gauge invariant. They capture the motion of dipoles along the $l$-direction. Inside the physical subspace we no longer have the $d_1$ operator and, hence, is not possible to build these line operators. The only gauge invariant line operators are along the time direction ($\exp i \int_t A_0^{(\alpha)}$), describing immobile excitations.

Next, we study the local operators that create excitations inside the physical subspace. Consider a generic local operator $e^{i\mathsf{T}KA(\mathbf{x}')}$, where $\mathsf{T}$ corresponds to an integer-valued vector to be determined under the condition that the resulting state lies inside the physical subspace. This is met provided

$$\mathcal{T}^{(1,\alpha)}K\mathsf{T} = 0. \tag{31}$$

In this way, the state

$$|\mathsf{T}\rangle_{\text{phy}} \equiv e^{i\mathsf{T}KA(\mathbf{x}')}|0\rangle_{\text{phy}} \tag{32}$$

is physical. Consider a general vector $\mathsf{T}$ with $\mathsf{T} = (n_1, n_2, n_3, n_4)$, with $n_a \in \mathbb{Z}$. The conditions in (31) imply

$$2n_1 + 6n_2 + 6n_3 = 0 \quad \text{and} \quad 2n_1 + 2n_2 + 4n_3 - 4n_4 = 0. \tag{33}$$

Thus, any operator characterized by a nontrivial vector of the form

$$\mathsf{T} = (-3n + 3m, -n - m, 2n, m), \quad n, m \in \mathbb{Z} \tag{34}$$

is responsible for creating excitations in the physical subspace. Let us construct them explicitly. To this end, we compute the commutator

$$
\begin{aligned}
\left[J_0^{(\alpha)}(\mathbf{x}), e^{i\mathsf{T}KA(\mathbf{x}')}\right] &= \sum_{I=1}^{2}(\mathcal{T}^{(I,\alpha)}K\mathsf{T})e^{i\mathsf{T}KA(\mathbf{x}')}d_I\,\delta(\mathbf{x}-\mathbf{x}') \\
&= (\mathcal{T}^{(2,\alpha)}K\mathsf{T})e^{i\mathsf{T}KA(\mathbf{x}')}d_2\,\delta(\mathbf{x}-\mathbf{x}').
\end{aligned}
\tag{35}
$$

Let us now determine the charge structure created by this local operator. By starting with a state $|0\rangle_{\text{phy}}$ with no charge content, i.e., $J_0^{(\alpha)}(\mathbf{x})|0\rangle_{\text{phy}} = 0$, the charge content of the state $|\mathsf{T}\rangle_{\text{phy}}$ is

$$J_0^{(\alpha)}(\mathbf{x})\,|\mathsf{T}\rangle_{\text{phy}} = q^{(\alpha)}d_2\,\delta(\mathbf{x}-\mathbf{x}')|\mathsf{T}\rangle_{\text{phy}}, \tag{36}$$

with the charges $q^{(\alpha)}$ defined as

$$q^{(\alpha)} \equiv \mathcal{T}^{(2,\alpha)}K\mathsf{T}. \tag{37}$$

For the two flavors of charges, one obtains

$$q^{(1)} = 4(n - m) \quad \text{and} \quad q^{(2)} = 4(m - 2n). \tag{38}$$

We see that operators characterized by vectors $\mathsf{T}$ with $m - 2n = 0$ create physical excitations of flavor $\alpha = 1$, while operators with $n - m = 0$ produce excitations of flavor $\alpha = 2$. Operators with both $n - m \neq 0$ and $m - 2n \neq 0$ produce excitations of the two flavors simultaneously.

### 3.3 Introducing Dynamics

In this section we consider dynamical terms in the the low-energy effective field theory (21) to study the spectrum of the excitations. We start by adding to the action an appropriate Maxwell-like term,

$$S = \int dt\, d^3x \left[\frac{1}{2g_E}F_{0a}F_{0a} + \frac{1}{4g_M}F_{ab}^{(\alpha)}F_{ab}^{(\alpha)} + \frac{1}{2\pi}K_{ab}A_a\,\partial_0 A_b + \frac{1}{\pi}A_0^{(\alpha)}K_{ab}\mathcal{D}_a^{(\alpha)}A_b\right], \tag{39}$$

where $F_{0a} \equiv \partial_0 A_a - \mathcal{D}_a^{(\alpha)}A_0^{(\alpha)}$, $F_{ab}^{(\alpha)} \equiv \mathcal{D}_a^{(\alpha)}A_b - \mathcal{D}_b^{(\alpha)}A_a$, and $g_E$ and $g_M$ are dimensionful couplings. We have chosen this particular form for the kinetic terms for simplicity, but in principle we could consider more general terms, for example, $F_{0a}M_{ab}F_{0b}$, involving an arbitrary matrix $M$.

A simple way to determine the spectrum of the excitations is to find the location of the poles of the propagator. To do this, we first fix the gauge $A_0^{(\alpha)} = 0$, which is allowed because of the existence of the two gauge degrees of freedom $\zeta^{(\alpha)}$. With this gauge choice, the equations of motion for the $A_a$ fields read

$$\left( -\frac{1}{g_E} \delta_{mb} \partial_0^2 + \frac{1}{g_M} \left[ \mathcal{D}_i^{(\alpha)} \mathcal{D}_i^{(\alpha)} \delta_{mb} - \mathcal{D}_m^{(\alpha)} \mathcal{D}_b^{(\alpha)} \right] + \frac{k}{\pi} \epsilon_{mb} \partial_0 \right) A_b = 0 . \tag{40}$$

Note that we have assumed an arbitrary level $k$ and re-wrote the $K$-matrix as $k \, \epsilon_{ab}$, where the $\epsilon_{ab}$ is a $4 \times 4$ anti-symmetric matrix with the elements in the upper triangle all equal to one. In momentum space, the equations of motion become

$$\left( \frac{1}{g_E} \delta_{mb} \omega^2 - \frac{1}{g_M} \left[ \mathcal{P}^2 \delta_{mb} - \mathcal{P}_m^{(\alpha)} \mathcal{P}_b^{(\alpha)} \right] + \frac{ik\omega}{\pi} \epsilon_{mb} \right) A_b = 0 , \tag{41}$$

where we have defined $\mathcal{P}_m^{(\alpha)} \equiv \mathcal{T}_m^{(I,\alpha)} p_I$, with $p_1 \equiv (p_x^2 + p_y^2 + p_z^2)/2$ and $p_2 \equiv (p_x p_y + p_x p_z + p_y p_z)$. The corresponding propagator obeys $\Delta_{mb} G_{bc} = \delta_{mc}$, i.e., $G_{bc} = \left( \Delta^{-1} \right)_{bc}$. The poles of $G_{bc}$ then follow from $\det(G) = 0$, which in general is a very complicated function of $\omega, p_x, p_y, p_z$. Since we are mostly interested in unveiling the gap for the excitations, we can focus on the limit $p_x, p_y, p_z \to 0$, where the determinant $\det(G)$ simplifies dramatically. In this limit the poles of $G$ can be obtained by solving the following quartic equation

$$\omega^4 - 6 \left( \frac{k}{\pi} \right)^2 g_E^2 \omega^2 + \left( \frac{k}{\pi} \right)^4 g_E^4 = 0 , \tag{42}$$

which leads to the following mass gaps

$$\omega_1 = \frac{k g_E}{\pi} \left( 1 + \sqrt{2} \right) \quad \text{and} \quad \omega_2 = \frac{k g_E}{\pi} \left( -1 + \sqrt{2} \right) . \tag{43}$$

In the limit where we have the pure Chern-Simons-like action, $g_E, g_M \to \infty$, the gap becomes infinitely large, which shows that the effective action (21) is fully gapped.

It is interesting to trace back the physical origin of the two independent excitations we have found. This comes from the number of physical components of the gauge fields, namely, we have two pairs of fields, with each pair giving rise to an independent excitation. This is a direct reflection of the number of degrees of freedom per site of the lattice model, where the local Hilbert space accommodates two spin-1/2 degrees of freedom. While the specific values obtained in (43) are tied to the choice $F_{0a} F_{0a}$, the fact that there are two excitations is independent of the form of the kinetic Maxwell-like term.

To further discuss this point, it is instructive to pursue a quantum mechanical analogy. The action (39) can be transformed into a simple quantum mechanical problem by studying field configurations depending only on time, $A_a \to \frac{1}{V^{1/2}} A_a(t)$, with $V$ the spatial volume of the system. With this, the action (39) reduces to

$$S = \int dt \left[ \frac{1}{2 g_E} \dot{A}_a^2 + \frac{k}{2\pi} \epsilon_{ab} A_a \dot{A}_b \right] . \tag{44}$$

With a simple change of basis we can split the action into two independent problems, with each one involving a pair of fields. This is achieved through an orthogonal transformation $A \to QA$, where the matrix $Q$ is given by

$$Q = \begin{pmatrix} 0 & -\frac{1}{\sqrt{2}} & 0 & \frac{1}{\sqrt{2}} \\ \frac{1+\sqrt{2}}{2\sqrt{3+2\sqrt{2}}} & -\frac{1}{2} & \frac{1-\sqrt{2}}{2\sqrt{3-2\sqrt{2}}} & -\frac{1}{2} \\ \frac{2+\sqrt{2}}{2\sqrt{3+2\sqrt{2}}} & 0 & \frac{2-\sqrt{2}}{2\sqrt{3-2\sqrt{2}}} & 0 \\ \frac{1+\sqrt{2}}{2\sqrt{3+2\sqrt{2}}} & \frac{1}{2} & \frac{1-\sqrt{2}}{2\sqrt{3-2\sqrt{2}}} & \frac{1}{2} \end{pmatrix} , \tag{45}$$

satisfying $QQ^\top = 1$ and $\det Q = 1$. The effect of this transformation in the action (44) is to leave the matrix $\epsilon$ in a block-diagonal form

$$Q\,\epsilon\,Q^\top = \begin{pmatrix} 0 & 1+\sqrt{2} & 0 & 0 \\ -(1+\sqrt{2}) & 0 & 0 & 0 \\ 0 & 0 & 0 & -1+\sqrt{2} \\ 0 & 0 & -(-1+\sqrt{2}) & 0 \end{pmatrix}. \tag{46}$$

In the new basis, the action (44) splits into two independent Landau problems with different magnetic fields,

$$S = S_1 + S_2, \tag{47}$$

where

$$S_1 \equiv \int dt \sum_{i=1}^{2} \left[ \frac{1}{2g_E}\dot{A}_i^2 + \frac{k(1+\sqrt{2})}{2\pi}\epsilon_{ij}A_i\dot{A}_j \right], \tag{48}$$

and

$$S_2 \equiv \int dt \sum_{i=1}^{2} \left[ \frac{1}{2g_E}\dot{A}_i^2 + \frac{k(-1+\sqrt{2})}{2\pi}\epsilon_{ij}A_i\dot{A}_j \right], \tag{49}$$

with $\epsilon_{ij}$ the two-dimensional antisymmetric index defined by $\epsilon_{12} \equiv 1$. The gap between the Landau levels is of the form $\omega = B/m$, where $B$ is the magnetic field and $m$ is the mass of the particle. From (48) and (49) we can identify $m_1 = m_2 \equiv g_E^{-1}$, $B_1 \equiv (1+\sqrt{2})k/\pi$ and $B_2 \equiv (-1+\sqrt{2})k/\pi$, so that the corresponding gaps between the Landau levels are

$$\omega_1 = \frac{g_E\,k}{\pi}\left(1+\sqrt{2}\right), \qquad \omega_2 = \frac{g_E\,k}{\pi}\left(-1+\sqrt{2}\right), \tag{50}$$

matching precisely the result obtained from the poles of the propagator. Therefore, we see that the effective field theory does capture properly the spectrum of low-lying excitations of the lattice model.

## 4  Discussions

We have been able to derive a low-energy effective field theory for the Haah code directly from the lattice model. This is done through a map that connects the spin operators of the lattice to field operators that are well-defined in the continuum limit. This procedure was developed in previous works in the context of type-I fractonic systems, like the X-cube model [36] and the Chamon code [35], and it is extended here to the case of a type-II fractonic system.

The effective theory is a 3+1 dimensional quadratic gauge theory of the Chern-Simons-type and hence fully gapped. The action is supplemented with a condition that is responsible for selecting physical states. This condition emerges from the representation of the identity in the continuum, which is not automatic when the lattice operators are represented in terms of field operators. Outside the physical subspace the theory contains mobile excitations and thus corresponds to a type-I fractonic theory. However, inside the physical subspace, all the excitations are completely immobile due to an infinite number of charge conservation laws that effectively takes place in this subspace. The type-II fractonic character is embedded into a type-I fractonic theory.

The physical properties inside the physical subspace are dictated by the operator $d_2 = \sum_{ijk} |\epsilon^{ijk}|\partial_j\partial_k$. In particular, the infinite set of conserved charges constructed out of functions that are annihilated by $d_2$ is an essential ingredient to ensure that all excitations are completely immobile. In this way, it is natural to ask whether it is possible to construct a

consistent effective theory involving a single derivative operator like $d_2$, without the need of any extra condition for selecting physical states. The action would be of the form (21) with $\mathcal{T}_a^{(1,\alpha)} = 0$ in (19) and $\mathcal{T}_a^{(2,\alpha)}$ satisfying $\mathcal{T}_a^{(2,\alpha)} K_{ab} \mathcal{T}_b^{(2,\beta)} = 0$ (for gauge invariance). This would be a pure type-II fracton theory, in the sense that it is not embedded into a type-I one. Reversing the steps to obtain the microscopic model from the effective theory is possible, although it does not lead to a consistent lattice model since the dictionary between $\Gamma^{(I,\alpha)}$ and $T^{(I,\alpha)}$ is not one-to-one, i.e., two distinct $T^{(I,\alpha)}$ are mapped into the same $\Gamma^{(I,\alpha)}$. Thus, it is possible to have a type-II fracton theory in the continuum with no obvious lattice model. The details can be found in the Appendix D. Whether there exists a lattice model for this pure type-II fracton continuum theory remains an open question.

## 5 Acknowledgements

We thank Guilherme Delfino for helpful discussions. This work is supported by the Brazilian agency Coordenação de Aperfeiçoamento de Pessoal de Nível Superior (CAPES) under grant number 88881.361635/2019-01 (W. F.), the CNPq grant number 311149/2017-0 (P. G.), and the DOE Grant No. DE-FG02-06ER46316 (C .C). W. F. acknowledges support by the Condensed Matter Theory Visitors Program at Boston University.

## A  Gauge Invariance

In the main text we comment that the gauge invariance property of the effective theory emerges from the fact that the lattice cubic operators commute among themselves. This property holds at any order of the expansion of cube operators, because it relies only on the $T$-vector structure. To see that, note that the differential operators can be written in general as

$$\mathcal{D}_a^{(\alpha)} = \sum_{I=0}^{7} T_a^{(I,\alpha)} \sum_j \frac{1}{j!} \left( \sum_{b=x,y,z} n_b^I \partial_b \right)^j, \tag{51}$$

where the vectors $n_b^I = \left( n_x^I, n_y^I, n_z^I \right)$ correspond to the corners of the cube. These positions are taken by considering that the origin sits at the center of the cube, namely,

$$n_b^1 = (-1,-1,-1), \quad n_b^2 = (-1,+1,-1), \quad n_b^3 = (+1,-1,-1), \quad n_b^4 = (-1,-1,+1),$$
$$n_b^5 = (+1,+1,-1), \quad n_b^6 = (-1,+1,+1), \quad n_b^7 = (+1,-1,+1), \quad n_b^8 = (+1,+1,+1). \tag{52}$$

This allows us to write the derivatives as

$$\mathcal{D}_a^{(1)} = \sum_j \frac{1}{j!} \Bigg[ -T_a^{(4,1)} \left( \left(-\partial_x + \partial_y - \partial_z\right)^j + \left(\partial_x - \partial_y - \partial_z\right)^j + \left(-\partial_x - \partial_y + \partial_z\right)^j \right)$$
$$\left( T_a^{(2,1)} + T_a^{(3,1)} \right) \left( \left(\partial_x + \partial_y - \partial_z\right)^j + \left(-\partial_x + \partial_y + \partial_z\right)^j + \left(\partial_x - \partial_y + \partial_z\right)^j \right) \tag{53}$$
$$- \left( T_a^{(1,1)} + T_a^{(5,1)} \right) \left(-\partial_x - \partial_y - \partial_z\right)^j \Bigg],$$

$$\mathcal{D}_a^{(2)} = \sum_j \frac{1}{j!} \Bigg[ T_a^{(5,2)} \left( \left(-\partial_x + \partial_y - \partial_z\right)^j + \left(\partial_x - \partial_y - \partial_z\right)^j + \left(-\partial_x - \partial_y + \partial_z\right)^j \right)$$
$$- \left( T_a^{(1,2)} + T_a^{(2,2)} \right) \left( \left(\partial_x + \partial_y - \partial_z\right)^j + \left(-\partial_x + \partial_y + \partial_z\right)^j + \left(\partial_x - \partial_y + \partial_z\right)^j \right) \tag{54}$$
$$- \left( T_a^{(3,2)} + T_a^{(4,2)} \right) \left(\partial_x + \partial_y + \partial_z\right)^j \Bigg].$$

We can put this in a more convenient form by using the identity

$$(x + y + z)^n = \sum_{k_1+k_2+k_3=n} \binom{n}{k_1,\ k_2,\ k_3} x^{k_1} y^{k_2} z^{k_3}\,. \tag{55}$$

Then the derivatives become

$$\mathcal{D}_a^{(1)} = \sum_j \sum_{k_1+k_2+k_3=j} \binom{j}{k_1,\ k_2,\ k_3} \frac{1}{j!}\bigg[-T_a^{(4,1)}\big((-1)^{k_1+k_2} + (-1)^{k_2+k_3} + (-1)^{k_1+k_3}\big)$$

$$\big(T_a^{(2,1)} + T_a^{(3,1)}\big)\big((-1)^{k_1} + (-1)^{k_2} + (-1)^{k_3}\big) - \big(T_a^{(1,1)} + T_a^{(5,1)}\big)(-1)^{k_1+k_2+k_3}\bigg]\partial_x^{k_1}\partial_y^{k_2}\partial_z^{k_3}\,, \tag{56}$$

$$\mathcal{D}_a^{(2)} = \sum_j \sum_{k_1+k_2+k_3=j} \binom{j}{k_1,\ k_2,\ k_3} \frac{1}{j!}\bigg[T_a^{(5,2)}\big((-1)^{k_1+k_2} + (-1)^{k_2+k_3} + (-1)^{k_1+k_3}\big)$$

$$-\big(T_a^{(1,2)} + T_a^{(2,2)}\big)\big((-1)^{k_1} + (-1)^{k_2} + (-1)^{k_3}\big) - \big(T_a^{(3,2)} + T_a^{(4,2)}\big)\bigg]\partial_x^{k_1}\partial_y^{k_2}\partial_z^{k_3}\,. \tag{57}$$

To proceed, we have to analyze the product $K_{ab}\mathcal{D}_a^{(\alpha)}\mathcal{D}_b^{(\beta)}$. The product involving operators of the same type, i.e., $\alpha = \beta$, automatically vanishes due to the anti-symmetry of the $K$ matrix. The product that can be in principle nonzero is the one involving $\mathcal{D}_a^{(1)}$ and $\mathcal{D}_b^{(2)}$. To analyze this, we can focus the attention only on the $T$-vector structure that arises from this product, which can be written as

$$-\left(\sum_{m=1}^{3}(-1)^{k_m}\right)\left(\sum_{n=1}^{3}(-1)^{q_n}\right)\big(T_a^{(2,1)} + T_a^{(3,1)}\big)K_{ab}\big(T_b^{(1,2)} + T_b^{(2,2)}\big)$$

$$-\left(\sum_{m=1}^{3}|\epsilon^{mij}|(-1)^{k_i+k_j}\right)\left(\sum_{n=1}^{3}|\epsilon^{npr}|(-1)^{q_p+q_r}\right)T_a^{(4,1)}K_{ab}T_b^{(5,2)}$$

$$+\left(\sum_{m=1}^{3}(-1)^{q_m+\sum_i k_i}\right)\big(T_a^{(1,1)} + T_a^{(5,1)}\big)K_{ab}\big(T_b^{(1,2)} + T_b^{(2,2)}\big)$$

$$+\left(\sum_{m=1}^{3}|\epsilon^{mij}|(-1)^{q_i+q_j}\right)T_a^{(4,1)}K_{ab}\big(T_b^{(3,2)} + T_b^{(4,2)}\big)$$

$$-\left(\sum_{m=1}^{3}(-1)^{k_m}\right)\big(T_a^{(2,1)} + T_a^{(3,1)}\big)K_{ab}\big(T_b^{(3,2)} + T_b^{(4,2)}\big)$$

$$-\left(\sum_{m=1}^{3}|\epsilon^{mij}|(-1)^{q_i+q_j+\sum_a k_a}\right)\big(T_a^{(1,1)} + T_a^{(5,1)}\big)K_{ab}T_b^{(5,2)}\,. \tag{58}$$

Now we have to check the situations where we have products between even (odd) differential operators. The parity of $\mathcal{D}$ is encoded in the sums $\sum_i k_i$ and $\sum_i q_i$. When we have the situation where both $\mathcal{D}$'s are even (odd) it amounts to analyze the cases in which $\sum_i k_i =$ even (odd) and $\sum_i q_i =$ even (odd). Let us analyze the even-even case, which can be achieved in two different ways: (i) $(k_1, k_2, k_3) =$ even; (ii) $(k_i, k_j) =$ odd, $i \neq j$ and $k_m =$ even, $m \neq i, j$ (similarly for the $q$'s). Therefore, the coefficients in (58) are such that the remaining combinations of the products between the $T$-vectors can be cancelled using the

commutation relations among the cubes $C^{(1)}$ and $C^{(2)}$,

$$\left(T_a^{(2,1)} + T_a^{(3,1)}\right) K_{ab} T_b^{(5,2)} = 0,$$

$$T_a^{(4,1)} K_{ab} \left(T_b^{(1,2)} + T_b^{(2,2)}\right) = 0,$$

$$\left(T_a^{(1,1)} + T_a^{(5,1)}\right) K_{ab} \left(T_b^{(3,2)} + T_b^{(4,2)}\right) = 0,$$

$$\left(T_a^{(2,1)} + T_a^{(3,1)}\right) K_{ab} \left(T_b^{(1,2)} + T_b^{(2,2)}\right) + T_a^{(4,1)} K_{ab} T_b^{(5,2)} = 0,$$

$$\left(T_a^{(1,1)} + T_a^{(5,1)}\right) K_{ab} \left(T_b^{(1,2)} + T_b^{(2,2)}\right) + T_a^{(4,1)} K_{ab} \left(T_b^{(3,2)} + T_b^{(4,2)}\right) = 0,$$

$$\left(T_b^{(2,1)} + T_b^{(3,1)}\right) K_{ab} \left(T_a^{(3,2)} + T^{(4,2)}\right) + \left(T_a^{(1,1)} + T_a^{(5,1)}\right) K_{ab} T_b^{(5,2)} = 0. \tag{59}$$

For completeness, we also introduce the commutation relations between cubes of the same type

$$T_a^{(4,1)} K_{ab} \left(T_b^{(2,1)} + T_b^{(3,1)}\right) = 0, \quad T_a^{(5,2)} K_{ab} \left(T_b^{(1,2)} + T_b^{(2,2)}\right) = 0,$$

$$T_a^{(4,1)} K_{ab} \left(T_b^{(1,1)} + T_b^{(5,1)}\right) = 0, \quad \left(T_a^{(3,2)} + T_a^{(4,2)}\right) K_{ab} T_b^{(5,2)} = 0,$$

$$\left(T_a^{(2,1)} + T_a^{(3,1)}\right) K_{ab} \left(T_b^{(1,1)} + T_b^{(5,1)}\right) = 0, \quad \left(T_a^{(1,2)} + T_a^{(2,2)}\right) K_{ab} \left(T_b^{(3,2)} + T_b^{(4,2)}\right) = 0. \tag{60}$$

By symmetry the odd-odd case is the same. Therefore, we obtain that for a theory with differential operators with the same parity, the condition that ensures gauge invariance under the transformations

$$A_a \rightarrow A_a + \mathcal{D}_a^{(\alpha)} \zeta^{(\alpha)},$$

$$A_0^{(\alpha)} \rightarrow A_0^{(\alpha)} + \partial_0 \zeta^{(\alpha)}, \tag{61}$$

is given by $K_{ab} \mathcal{D}_a^{(\alpha)} \mathcal{D}_b^{(\beta)} = 0$.

A similar relation can be obtained when the differential operators possess different parities, i.e., one derivative is odd and the other is even. The difference now is that the coefficients in (58) of the odd derivatives will acquire a relative sign, such that (59) can no longer be used to cancel all the terms in (58). This is because the gauge invariance for these theories with derivatives possessing different parities demands that we introduce an operator $\bar{\mathcal{D}}_a^{(\alpha)}$, which differs from $\mathcal{D}_a^{(\alpha)}$ by a minus sign on every odd term. For these theories, the gauge invariance under the transformations

$$A_a \rightarrow A_a + \bar{\mathcal{D}}_a^{(\alpha)} \zeta^{(\alpha)},$$

$$A_0^{(\alpha)} \rightarrow A_0^{(\alpha)} + \partial_0 \zeta^{(\alpha)}, \tag{62}$$

follows from the condition $K_{ab} \mathcal{D}_a^{(\alpha)} \bar{\mathcal{D}}_b^{(\beta)} = 0$, which is ensured by (59) once again.

# B  Mobility in the Unconstrained Model

To see that the model with no constraint in the Hilbert space supports mobile excitations it is convenient to introduce the set of orthonormal coordinates $(l, u, v)$, defined as

$$\hat{l} \equiv \frac{1}{\sqrt{3}} \left(\hat{x} + \hat{y} + \hat{z}\right), \quad \hat{u} \equiv \frac{1}{\sqrt{2}} \left(\hat{y} - \hat{z}\right), \quad \hat{v} \equiv \frac{1}{\sqrt{6}} \left(-2\hat{x} + \hat{y} + \hat{z}\right). \tag{63}$$

In terms of these new coordinates, the differential operators $\mathcal{D}_a^{(\alpha)}$ become

$$\mathcal{D}_a^{(\alpha)} = T_a^{(1,\alpha)} \underbrace{\frac{1}{2} \left(\partial_l^2 + D_{uv}\right)}_{d_1} + T_a^{(2,\alpha)} \underbrace{\left(\partial_l^2 - \frac{1}{2} D_{uv}\right)}_{d_2}, \quad D_{uv} \equiv \partial_u^2 + \partial_v^2, \tag{64}$$

which makes evident the underlying symmetries. We have the conservation of two global charges

$$Q^{(\alpha)} \equiv \int dl \, du \, dv \, J_0^{(\alpha)}. \tag{65}$$

Moreover, we have an infinite number of conserved charges, which are responsible for certain mobility restrictions on the excitations. Indeed, the charges

$$Q_f^{(\alpha)} \equiv \int dl \, du \, dv \, f(l, u, v) J_0^{(\alpha)} \tag{66}$$

are conserved provided that $f(l, u, v)$ satisfies $\partial_l^2 f = D_{uv} f = 0$. The most general solution of these conditions is $f(l, u, v) = (c_1 l + c_2) h(u, v)$, where $c_1$ and $c_2$ are arbitrary constants and $h(u, v)$ is a harmonic function in the $u - v$ plane, i.e., $D_{uv} h(u, v) = 0$. Now consider, for example, the density corresponding to a point charge $q^{(\alpha)}$ located at $(u_0(t), v_0(t), l_0(t))$, at the time $t$, $J_0^{(\alpha)}(t, u, v, l) = q^{(\alpha)} \delta(u - u_0(t)) \delta(v - v_0(t)) \delta(l - l_0(t))$. Using this in (66) gives

$$Q_f^{(\alpha)} = q^{(\alpha)}(c_1 l_0(t) + c_2) h(u_0(t), v_0(t)), \tag{67}$$

which is conserved only if $\frac{du_0}{dt} = \frac{dv_0}{dt} = \frac{dl_0}{dt} = 0$. In other words, conservation of this infinite set of charges enforces all single-particle excitations to be completely immobile. However, we can still have mobility along the $l$-direction of higher order multipoles, such as dipoles, quadrupoles, etc. Consider the density with charges $q^{(\alpha)}$ and $-q^{(\alpha)}$ located at the positions $u_0, v_0, l_0$ and $u_1, v_1, l_1$:

$$\begin{aligned} J_0^{(\alpha)}(t, u, v, l) = q^{(\alpha)} \big[ & \delta(u - u_0(t)) \delta(v - v_0(t)) \delta(l - l_0(t)) \\ & - \delta(u - u_1(t)) \delta(v - v_1(t)) \delta(l - l_1(t)) \big]. \end{aligned} \tag{68}$$

This implies

$$Q_f^{(\alpha)} = q^{(\alpha)}(c_1 l_0(t) + c_2) h(u_0(t), v_0(t)) - q^{(\alpha)}(c_1 l_1(t) + c_2) h(u_1(t), v_1(t)). \tag{69}$$

Thus we see that if we set $u_1 = u_0 = \text{const}$, $v_1 = v_0 = \text{const}$ and $l_0 - l_1 = \text{const}$, the above charge will be conserved. This configuration corresponds to the motion of a dipole disposed along the $l$-direction moving in parallel to its axis. In this sense, without the projection onto the physical subspace (17), the enlarged effective field theory (21) describes type-I fractons.

## C   Higher Moment Conservations and the Associated Polynomials

Consider a general polynomial in three variables, $(x, y, z)$, of degree $d$ written in terms of a monomial basis $\mathbb{R}_d(x, y, z)$

$$P^{(d)} = \sum_\alpha p_\alpha X^\alpha, \tag{70}$$

where $p_\alpha = p_{abc}$ and $X^\alpha = x^a y^b z^c$ and $\alpha$ runs over all the elements that belong to $\mathbb{R}_d(x, y, z)$. In the following, we omit the dependence on the coordinates $(x, y, z)$. The basis $\mathbb{R}_d$ has dimension

$$\kappa_d = \dim(\mathbb{R}_d) = \frac{(d+2)!}{d! \, 2!}. \tag{71}$$

In the main text, we have argued that there are infinitely many conserved charges that follow from the polynomial solutions $P^{(d)}$ of the equation $d_2 P^{(d)} = 0$, with

$d_2 = \partial_x \partial_y + \partial_x \partial_z + \partial_y \partial_z$ realizing the mapping $d_2 : \mathbb{R}_d \to \mathbb{R}_{d-2}$. Then our problem amounts to finding the kernel of the differential operator $d_2$. The dimension of the kernel can be easily found by counting the number of free coefficients after solving $d_2 P^{(d)} = 0$. Therefore, the dimension of the kernel is

$$\dim(\ker(d_2)) = \kappa_d - \kappa_{d-2} = \frac{(d+2)(d+1) - d(d-1)}{2}. \tag{72}$$

The elements of the monomial basis $\mathbb{R}_d$ can be represented as

$$\mathbb{R}_d = \left\{ x^d, x^{d-1}y, x^{d-1}z, x^{d-2}y^2, x^{d-2}yz, x^{d-2}z^2, \ldots y^d, y^{d-1}z, y^{d-2}z^2, \ldots, z^d \right\}, \tag{73}$$

and we associate a coefficient $p_\alpha$ for each of the elements above, i.e., $p_1$ is associated with the element in the first entry, $p_2$ is associated with the element at the second entry, and so on. It is immediate to note that any polynomial built from $\mathbb{R}_0$ and $\mathbb{R}_1$ trivially satisfies $d_2 P^{(d)} = 0$ and the dimension of the kernels are 1 and 3, respectively. The first nontrivial case happens for polynomials built from $\mathbb{R}_2$ with the generic form

$$P^{(2)} = p_1 x^2 + p_2 xy + p_3 xz + p_4 y^2 + p_5 yz + p_6 z^2. \tag{74}$$

Solving $d_2 P^{(2)} = 0$ fixes $p_2 = -p_3 - p_5$, and therefore a solution is the polynomial

$$P^{(2)} = p_1 x^2 + p_3 (xz - xy) + p_4 y^2 + p_5 (yz - xy) + p_6 z^2. \tag{75}$$

This polynomial is a solution for any $p_1, p_3, p_4, p_5, p_6$ and will lead to the conservation of some component (or combination of components) of the quadrupole moment. The same analysis holds for higher degrees. In the following we write some solutions of higher orders,

$$P^{(3)} = \sum_{\alpha \notin \pi^{(3)}} p_\alpha X^\alpha + (p_6 - p_3 + p_9) x^2 y + (p_6 - p_8 + p_9) xy^2 - (2p_6 + 2p_9) xyz, \tag{76}$$

$$P^{(4)} = \sum_{\alpha \notin \pi^{(4)}} p_\alpha X^\alpha + \left( \frac{2p_6}{3} - p_3 - p_{10} - p_{14} \right) x^3 y + (p_6 - 3p_{10} + p_{13} - 3p_{14}) x^2 y^2$$
$$+ (3p_{10} + 3p_{14} - 2p_6) x^2 yz + \left( \frac{2p_{13}}{3} - p_{10} - p_{12} - p_{14} \right) xy^3$$
$$+ (3p_{10} - 2p_{13} + 3P_{14}) xy^2 z - (3p_{10} - 3p_{14}) xyz^2, \tag{77}$$

$$P^{(5)} = \sum_{\alpha \notin \pi^{(5)}} p_\alpha X^\alpha + \left( \frac{p_6}{2} - p_3 - \frac{p_{10}}{2} + p_{15} + p_{20} \right) x^4 y \tag{78}$$
$$+ (p_6 - 2p_{10} + 6p_{15} - p_{19} + 6p_{20}) x^3 y^2$$
$$+ (2p_{10} - 2p_6 - 4p_{15} - 4p_{20}) x^3 yz + (6p_{15} - p_{10} + p_{18} - 2p_{19} + 6p_{20}) x^2 y^3$$
$$+ (3p_{10} - 12p_{15} + 3p_{19} - 12p_{20}) x^2 y^2 z + (6p_{15} - 3p_{10} + 6p_{20}) x^2 yz^2$$
$$+ \left( p_{15} - p_{17} + \frac{p_{18}}{2} - \frac{p_{19}}{2} + p_{20} \right) xy^4 + (2p_{19} - 4p_{15} - 2p_{18} - 4p_{20}) xy^3 z$$
$$+ (6p_{15} - 3p_{19} + 6p_{20}) xy^2 z^2 - (4p_{15} + 4p_{20}) xyz^3, \tag{79}$$

$$P^{(6)} = \sum_{\alpha \notin \pi^{(6)}} p_\alpha X^\alpha + \left( \frac{2p_6}{5} - p_3 - \frac{3p_{10}}{10} + \frac{2p_{15}}{5} - p_{21} - p_{27} \right) x^5 y$$

$$+ \left( p_6 - \frac{3p_{10}}{2} + 3p_{15} - 10p_{21} + p_{26} - 10p_{27} \right) x^4 y^2 \tag{80}$$

$$+ \left( \frac{3p_{10}}{2} - 2p_6 - 2p_{15} + 5p_{21} + 5p_{27} \right) x^4 y z$$

$$+ \left( 4p_{15} - p_{10} - 20p_{21} - p_{25} + 4p_{26} - 20p_{27} \right) x^3 y^3 \tag{81}$$

$$+ \left( 3p_{10} - 8p_{15} + 30p_{21} - 4p_{26} + 30p_{27} \right) x^3 y^2 z$$

$$+ \left( 4p_{15} - 3p_{10} - 10p_{21} - 10p_{27} \right) x^3 y z^2 \tag{82}$$

$$+ \left( p_{15} - 10p_{21} + p_{24} - \frac{3p_{25}}{2} + 3p_{26} - 10p_{27} \right) x^2 y^4$$

$$+ \left( 30p_{21} - 4p_{15} + 3p_{25} - 8p_{26} + 30p_{27} \right) x^2 y^3 z \tag{83}$$

$$+ \left( 6p_{15} - 30p_{21} + 6p_{26} - 30p_{27} \right) x^2 y^2 z^2$$

$$+ \left( 10p_{21} - 4p_{15} + 10p_{27} \right) x^2 y z^3 + \left( \frac{2p_{24}}{5} - p_{21} - p_{23} - \frac{3p_{25}}{10} + \frac{2p_{26}}{5} - p_{27} \right) x y^5$$

$$+ \left( 5p_{21} - 2p_{24} + \frac{3p_{25}}{2} - 2p_{26} + 5p_{27} \right) x y^4 z + \left( 4p_{26} - 10p_{21} - 3p_{25} - 10p_{27} \right) x y^3 z^2$$

$$+ \left( 10p_{21} - 4p_{26} + 10p_{27} \right) x y^2 z^3 - \left( 5p_{21} + 5p_{27} \right) x y z^4, \tag{84}$$

where $\pi^{(3)} = \{p_2, p_4, p_5\}$, $\pi^{(4)} = \pi^{(3)} \cup \{p_7, p_8, p_9\}$, $\pi^{(5)} = \pi^{(4)} \cup \{p_{11}, p_{12}, p_{13}, p_{14}\}$, $\pi^{(6)} = \pi^{(5)} \cup \{p_{16}, p_{17}, p_{18}, p_{19}, p_{20}\}$. This construction continues to polynomials of higher degrees. In each step the number of linearly dependent coefficients increases with the degree of the previous polynomial, i.e., for a polynomial of degree seven the coefficients contained in $\pi^{(7)} = \pi^{(6)} \cup \{p_{22}, p_{23}, p_{24}, p_{25}, p_{26}, p_{27}\}$ will be given in terms of all others that are not contained in $\pi^{(7)}$. In general, all linear dependent coefficients $p_i$ will be contained in the set

$$\pi^{(d)} = \pi^{(d-1)} \cup \{p_{i+2}, p_{i+3}, \ldots, p_{i+d}\}, \quad d > 2, \tag{85}$$

where $p_i$ is the coefficient that sits at the last entry of $\pi^{(d-1)}$. We are assuming that one organizes these coefficients in increasing order. Note that the set represented by the curly brackets contains $(d-1)$ elements.

The conclusion is that the effective theory for the Haah code contains infinitely many conserved charges, and these charges can be interpreted as the components of higher moment multipoles (dipoles, quadrupoles, octupoles and so on) or combinations thereof.

## D  A type-II Continuum Fracton Model Without an Apparent Lattice Counterpart

In this section we expand the discussion in the main text about constructing a type-II fracton field theory involving exclusively the $d_2$ differential operator. We emphasize that we will change the notation slightly. The corners of the cubes will be labeled from 0 to 7. As before, the product of Dirac operators in a given corner is determined by a vector $T_a^{(I,\alpha)}$, $I = 0, \ldots, 7$ according to (9). We will consider consider two cube operators similarly to Fig.(1), but with arbitrary Dirac operators in each of the corners. Upon expansion of these cube operators (like the one that led to (14)) one can demand that the coefficients of the differential operators $\mathbb{1}, \partial_i, \partial_i^2$ vanish identically and that the coefficients of $\partial_i \partial_j$, $i \neq j$ are such that the resulting differential operator is $\mathcal{D}_a^{(\alpha)} = \mathcal{T}_a^{(\alpha)} d_2$. These requirements give us a set of equations that allow

us to determine the allowed $T$-vectors and hence the Dirac operators. However, as we shall argue below, there are some ambiguities arising in returning from the continuum to the lattice which render the lattice model inconsistent with the continuum one. Although we are able to construct a continuum theory compatible with the physics of a pure type-II fracton model, it seems very difficult to obtain a corresponding lattice model within the framework discussed here.

To make this problem explicit, let us consider a generic cube operator

$$C_{\mathbf{x}}^{(\alpha)} = \exp\left( i \sum_{I=0}^{7} T_a^{(I,\alpha)} K_{ab} A_b \left( \mathbf{x} + \hat{r} \cdot \mathbf{n}_I \right) \right), \tag{86}$$

where $\hat{r} = (\hat{x}, \hat{y}, \hat{z})$ and $\mathbf{n}_I$ is a set of vectors specifying the positions of the corners of the cube relative to its center, explicitly given by

$$\mathbf{n}_0 = -(1, 1, 1), \quad \mathbf{n}_1 = (-1, -1, 1), \quad \mathbf{n}_2 = (-1, 1, -1), \quad \mathbf{n}_3 = (-1, 1, 1),$$
$$\mathbf{n}_4 = (1, -1, -1), \quad \mathbf{n}_5 = (1, -1, 1), \quad \mathbf{n}_6 = (1, 1, -1), \quad \mathbf{n}_7 = (1, 1, 1).$$

We shall impose constraints on the combinations of $T$-vectors emerging from the expansion so that the continuum effective theory contains the single differential operator $d_2 = \sum_{ijk} |\epsilon^{ijk}| \partial_j \partial_k$. Such theory is a candidate for a pure type-II fracton. We shall determine the corresponding $T$-vectors and then try to reconstruct the lattice model.

Expanding the cube operators in (86) leads to a continuum theory. The resulting terms in the exponential can be organized according to the order of the derivatives they involve, namely, $\mathbb{1}$, $\partial_i$, $\partial_i^2$, and $\partial_i \partial_j$, with $i \neq j$. The coefficients of each one of these differential operators are linear combinations of the $T$-vectors. We then demand that the coefficients of $\mathbb{1}$, $\partial_i$, and $\partial_i^2$ vanish. In addition, we impose that the coefficients of the operators $\partial_i \partial_j$, with $i \neq j$, are the same in order to identify the operator $d_2$. These conditions lead to a system of equations that can be solved. We find that only two $T$-vectors in each cube operator are independent, say, $T^{(6,\alpha)}$ and $T^{(7,\alpha)}$. The remaining ones can be written in terms of these two vectors:

$$\begin{aligned} T^{(0,\alpha)} &= 3T^{(6,\alpha)} + 2T^{(7,\alpha)}, \\ T^{(1,\alpha)} &= T^{(2,\alpha)} = T^{(4,\alpha)} = -2T^{(6,\alpha)} - T^{(7,\alpha)}, \\ T^{(3,\alpha)} &= T^{(5,\alpha)} = T^{(6,\alpha)}. \end{aligned} \tag{87}$$

The cube operators in (86) reduce to

$$C_{\mathbf{x}}^{(\alpha)} \sim \exp\left( i K_{ab} \mathcal{D}_a^{(\alpha)} A_b(\mathbf{x}) \right), \tag{88}$$

where now $\mathcal{D}_a^{(\alpha)} \equiv \mathcal{T}_a^{(\alpha)} d_2$, with $\mathcal{T}^{(\alpha)} \equiv 4 \left( T^{(6,\alpha)} + T^{(7,\alpha)} \right)$. We can immediately write down the corresponding effective action in the form (21), which is invariant under gauge transformations $A_a \rightarrow A_a + \mathcal{D}_a^{(\alpha)} \zeta^{(\alpha)}$, provided that $K_{ab} \mathcal{D}_a^{(\alpha)} \mathcal{D}_b^{(\beta)} = 0$. This condition is equivalent to $\mathcal{T}_a^{(\alpha)} K_{ab} \mathcal{T}_b^{(\beta)} = 0$. Therefore, gauge invariance of the continuum theory imposes restrictions only on $\mathcal{T}^{(\alpha)}$, but not on $T^{(6,\alpha)}$ and $T^{(7,\alpha)}$ individually. We refer to this condition as the *weak gauge condition*.

With different discretization prescriptions for $d_2 \delta(\mathbf{x} - \mathbf{x}')$ in the expression for charge density in the relation (27) but with $\mathcal{D}_a^{(\alpha)} \equiv \mathcal{T}_a^{(\alpha)} d_2$, we can glimpse several charge arrangements that would be present in the corresponding lattice model. For example, if we consider $\partial_i f(\mathbf{x}) \rightarrow f(\mathbf{x} + \epsilon_i) - f(\mathbf{x})$ in terms of the coordinates $l$, $u$, $v$, we obtain the configuration (a) in Fig. 2, since in these coordinates $d_2 = \partial_l^2 - (\partial_u^2 + \partial_v^2)/2$. On the other hand, using the same discretization above, but in terms of coordinates $x$, $y$, $z$, we obtain the configuration depicted in (b) of Fig. 2, since $d_2 = \partial_x \partial_y + \partial_y \partial_z + \partial_x \partial_z$. It is important to stress that since the

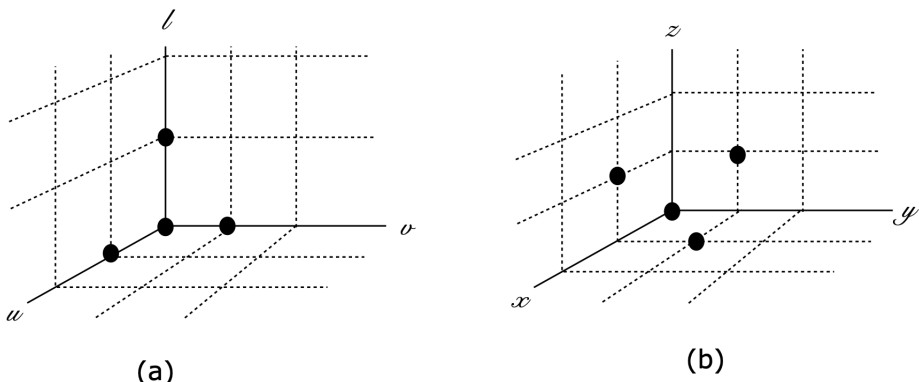

Figure 2: Charge distributions created by the operator $d_2$. In configuration (a) we set the lattice spacing as $\epsilon_u = \epsilon_v = \epsilon_l/\sqrt{2}$, whereas in (b) we are considering equal lattice spacing.

charge $q^{(\alpha)}$ is only defined mod $2q^{(\alpha)}$, the black dots in Fig. 2 represents the positions of an odd number of charges, since $(2\mathbb{Z}+1)q^{(\alpha)} \sim q^{(\alpha)}$ and $2\mathbb{Z}q^{(\alpha)} \sim 0$. Naturally, the discretization procedure is not unique, since from the continuum perspective there is no *a priori* preferable way of discretizing derivatives.

Let us try to reconstruct the lattice model. The main issue with the weak gauge condition is that it likely will lead to a lattice model that is not given in terms of commuting projectors, i.e., a lattice model where the cube operators are noncommuting. To obtain a lattice model of commuting projectors we need some lattice input. We import the commutation relations from the lattice, which correspond to restrictions on the vectors $T^{(6,\alpha)}$ and $T^{(7,\alpha)}$ individually, namely,

$$
\begin{aligned}
T_a^{(6,1)} K_{ab} T_b^{(7,1)} &= 0\,, & T_a^{(6,2)} K_{ab} T_b^{(7,2)} &= 0\,, \\
T_a^{(6,1)} K_{ab} T_b^{(7,2)} &= 0\,, & T_a^{(7,1)} K_{ab} T_b^{(6,2)} &= 0\,, \\
T_a^{(6,1)} K_{ab} T_b^{(6,2)} + T_a^{(7,1)} K_{ab} T_b^{(7,2)} &= 0\,.
\end{aligned}
\tag{89}
$$

These conditions are stronger than the previous ones and, consequently, also ensure gauge invariance of the continuum theory. Moreover, they ensure that all the cube operators are simultaneously commuting. An explicit solution for this set is $T^{(6,1)} = (0,0,0,1)$, $T^{(7,1)} = (0,-1,1,0)$, $T^{(6,2)} = (1,1,-1,1)$, and $T^{(7,2)} = (1,-1,0,2)$. This enables us to identify the following spin operators

$$
T^{(6,1)} \rightarrow \gamma^4\,, \quad T^{(7,1)} \rightarrow \gamma^2\gamma^3\,, \quad T^{(6,2)} \rightarrow \gamma^5\,, \quad T^{(7,2)} \rightarrow \gamma^1\gamma^2\,.
\tag{90}
$$

The corresponding cube operators are shown in Fig.(3). The lattice theory built from those cubes is *not* compatible with the effective theory constructed from the operator (88). In fact, the lattice model defined in terms of the cubes of Fig. (3) supports *mobile excitations*, corresponding to a type-I fracton system. The mobility can be seen from the fact that one can act with a local operator that anti-commutes with all operators in $C^{(1)}$ (or $C^{(2)}$), and thus create eight defects, an octupole configuration. For example, take the action of a $\gamma_2$ operator on a single site, since $\gamma_2$ anti-commutes with all operators in $C^{(1)}$ (and $C^{(2)}$ as well in this case), it will change the sign of eight neighboring cubes, thus creating an ocutpole excitation. The octupoles can then be used to move quadrupole excitations.

So what goes wrong? The subtle point is that there are ambiguities in the lattice which are not innocuous in the continuum theory. We have come across with this before: the identity operator in the lattice is not automatic implemented in the continuum theory. Let

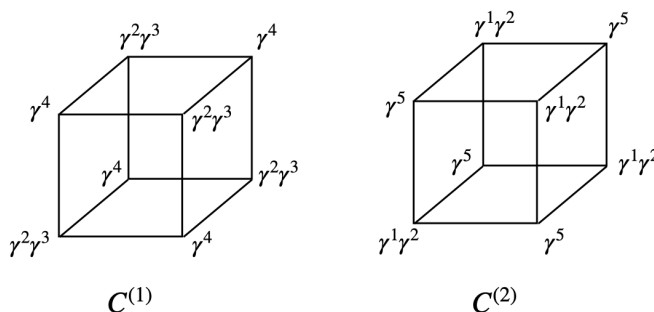

Figure 3: Cubes obtained from the effective field theory.

us consider another example, say, two operators characterized respectively by $T^{(6,1)}$ and $T^{(0,1)} = 3T^{(6,1)} + 2T^{(7,1)}$. While they are distinct from the continuum point of view, they lead to the same lattice operator $\gamma^4$, since the components of the $T$-vectors are defined mod 2. This violates the one-to-one map between $T^{(I,\alpha)}$ and the lattice operators, since two $T$-vectors are mapped to the same operator. Therefore, this lattice model is not a valid one. This can be made more explicitly if one starts with the lattice model defined by the cubes of Fig.(3) and applies the procedure in the main text to obtain the effective theory. The resulting theory corresponds to a fracton system with mobile quadrupole excitations, representing properly the lattice model, and not a type-II theory as the one that follows from (88). While the passage from the lattice to the continuum using the framework described here and in [35] safely produces a *bona fide* effective description, the reverse is not true. On the other hand, the effective action that follows from (88) corresponds to a properly pure type-II continuum fractonic theory, but with no any obvious corresponding lattice model.

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
