# Peer review of "Field Theories for type-II fractons"

_SciPost Physics, doi:SciPost Phys. 12, 064 (2022)_

## Round 2 · Referee Report · Anonymous (Referee 1) · 2021-6-5

Strengths

This work constructs an effective field theory for the Haah code, a prototypical example of type-II fracton models. It makes an interesting proposal that in order to capture the physics of the lattice model, the effective field theory has to be supplemented with an additional constraint that projects the Hilbert space to certain subspace. Without imposing the constraint, the effective field theory includes mobile dipoles and hence cannot describe a type-II fracton model.

Weaknesses

  1. It is not clear that after restricting the Hilbert space, the effective field theory is a consistent theory.
  2. The global aspects of the effective field theory are not discussed.

Report

This work proposes an interesting idea on the important questions of constructing effective field theories for fracton model. However, I have a few concerns/questions/comments that I hope the authors could address before this work is published.

1. Is the method used in this work applicable only to $\mathbb{Z}_2$ theories? Can it be generalized to $\mathbb{Z}_N$ theories, such as $\mathbb{Z}_N$ X-cube model or Haah code? It might be worth adding a few sentences commenting on this issue.

2. Is the equation (2) and (3) correct? Could there be an overall minus sign missing? If so, does it affect the analysis afterward?

3. Do the commutation relations in equation (6) reproduce $(\gamma^I)^2=1$?

4. In equation (12), should it be $d_1=\frac{1}{2}(\partial_x^2+\partial_y^2+\partial_z^2)$?

5. Is the condition in equation (14) gauge invariant? If so, it might be worth stressing this. I am also concerned about the consistency of the theory on this restricted Hilbert space. Usually, given a consistent theory, naively restricting the Hilbert space to a subspace leads to an inconsistent theory.

6. If I am not mistaken, the Haah code should have two types of excitations for each flavor label $\alpha$. For example, we can create excitations that only excite $C^{(1)}$ using $Z$ operator that can either act on the first spin or the second spin on a site. I assume that the charge configurations in figure 2(a) and 2(b) are related by coordinate transformation. Please correct me if my understanding is incorrect. If so, then the continuum field theory seems to only capture one type of excitations as shown in figure 2(b). What happens to the other excitations?

7. In (A.4) should the last term in $\mathcal{D}_a^{(2)}$ be $(\partial_x+\partial_y+\partial_z)^j$ instead of $(-\partial_x-\partial_y-\partial_z)^j$?

8. In (A.9) should the second equation be $T_a^{(4,1)}K_{ab}\left(T_b^{(1,2)}+ T_b^{(2,2)}\right)=0$ instead?

9. In (B2) should $D_{uv}=\partial_u^2+3\partial_v^2$?

Requested changes

I hope the authors could make appropriate changes in the draft addressing some of questions I raised above. In addition to the questions above, I would also suggest the following changes:

Major changes:

1. Various global aspects of the action in equation (18) is not discussed. For example, is the gauge field compact or non-compact? If the gauge field is compact, what is the periodicity and how it affects the operator spectrum? The authors should acknowledge that the global properties of the effective field theory are not discussed.

2. In the last paragraph of page 9, the author claims that there exist line operators in the full Hilbert space. It would help the reader if the line operators were written down explicitly.

3. It is not clear to me how the differential operator $d_2$ is discretized to obtained the charge arrangement in figure (2). It would help if the authors could elaborate on this procedure, and clarify whether this procedure has any ambiguity or not.

4. In the paragraph below (D5), the author claims that there is a local operator that can create eight defects. It would help the readers if the authors can write down which local operator explicitly.

Minor changes:

1. In the first paragraph of introduction, “This dependence on the lattice details signals a sort of infrared/ultraviolet (IR/UV) mixing …”. I believe the standard terminology is “UV/IR mixing”.

2. In figure 1, add $\dagger$ on the appropriate $\gamma$ operators as in equation (2) and (3).

3. On page 5, “… can be read sistematically from …” -> “… can be read systematically from …”.

4. In the paragraph below (14), $q^{(\alpha)} \mod 2q^{(\alpha)}$ -> $q^{(\alpha)} \mod 2$.

5. In the second paragraph of page 9, the variables $l, u, v$ are first mentioned but they are not defined.

6. Below (A12), “diferent” -> “different”.

7. In appendix D, $T^{I,\alpha}_a$ has index $I$ ranging from 0 to 7 instead of 1 to 5 as in the main text. This might confuse the reader. Also, it is not explained how the index $I$ and vector $n_I$ are labeled around a cube.

  • validity: good
  • significance: high
  • originality: high
  • clarity: ok
  • formatting: good
  • grammar: good

Author:  Weslei Fontana  on 2021-10-19  [id 1865]

(in reply to Report 1 on 2021-06-05)
Category:
answer to question

Referee

Is the method used in this work applicable only to $Z_2$ theories? Can it be generalized to $Z_N$ theories, such as $Z_N$ X-cube model or Haah code? It might be worth adding a few sentences commenting on this issue

response We believe that in principle the method can be generalized to $Z_N$ theories. More specifically, we have discussed the $Z_N$ case for a type-I fracton model in the appendix E of SciPost Phys. Core 4, 012 (2021). This generalization corresponds to different choices of the parameter $k$ entering the $K$-matrix (the $Z_2$ case corresponds to $k=1$), and this can be established by considering the ground state degeneracy coming from the effective field theory through the study of the algebra of the line operators. However, for the case of the Haah code, the situation is far more complicated, since the ground state degeneracy is unknown (also there are no line operators). Nevertheless, the effective field theory for the Haah code does make sense for a generic parameter $k$, as we did in the new section (IIIC) introduced in the revised manuscript, but it is not clear if it realizes the $Z_{N}$ symmetry.

referee

Is the equation (2) and (3) correct? Could there be an overall minus sign missing? If so, does it affect the analysis afterward?

response Indeed, taking into account the appropriate factors of $i$ in the $\gamma$ operators there is an overall minus sign missing in the cubes $C^{(i)}$. However, this sign is innocuous in the analysis since the physics of the Haah code is insensitive to an overall minus sign in the Hamiltonian. The overall sign can be changed by a gauge transformation acting on the degrees of freedom on one of the sublattices. For example, by rotating $X\rightarrow -X$ in only the second spin in only one sublattice (either the even or odd), one obtains that $C^{(1)}\rightarrow -C^{(1)}$. Similarly, one can change the sign of the $C^{(2)}$ cube. We remark that this freedom of sign choice for the coupling present in the Haah code is not general, for example one cannot change all the signs of the star terms in the X-cube model with a gauge transformation.

referee

Do the commutation relations in equation (6) reproduce $(\gamma^I)^2 = 1$ ?

response The commutation relations in (6) imply that the representation $\gamma_{\vec{x}}^{(I)}=\exp [i t_a^{(I)}K_{ab}A_b(\vec{x})]$ satisfies $[(\gamma^I)^2,\gamma^J]=0$ for any $I$ and $J$, i.e. $(\gamma^I)^2=\exp[2 i t_a^{(I)}K_{ab}A_b(\vec{x})]$ acts as the identity within commutator.

referee

Is the condition in equation (14) gauge invariant? If so, it might be worth stressing this. I am also concerned about the consistency of the theory on this restricted Hilbert space. Usually, given a consistent theory, naively restricting the Hilbert space to a subspace leads to an inconsistent theory.

response The selection rule (17) [previously Eq. (14) in the original version] is not gauge invariant. In general, it does not define consistently a subspace which is closed under time evolution, since it does not commute with the full Hamiltonian. However, we use it only after the theory is properly projected onto the ground state (in the low-energy effective theory), where the closure under time evolution is trivial. We added this comment in the paragraph below Eq. (17).

referee

If I am not mistaken, the Haah code should have two types of excitations for each flavor label $\alpha$. For example, we can create excitations that only excite $C^{(1)}$ using $Z$ operator that can either act on the first spin or the second spin on a site. I assume that the charge configurations in Figure 2(a) and 2(b) are related by coordinate transformation. Please correct me if my understanding is incorrect. If so, then the continuum field theory seems to only capture one type of excitations as shown in Figure 2(b). What happens to the other excitations?

response It is correct that the Haah code has two type of excitations for each flavor $\alpha$ and that the configurations in Fig. 2(a) and 2(b) are related by coordinate transformations as we take the continuum limit (they correspond to distinct discretizations with different lattice parameters). After considering carefully this question, we have concluded that the discretization discussed at the end of the Sec. III of the original version of the manuscript is naive. The basic reason is that we have discretized it after taking into account the condition (17). On the other hand, this very condition only emerged because we took the continuum limit, i.e., it does not need to be enforced in the lattice model since the corners of the cube operators are not collapsed to a single point. Nevertheless, this reasoning is legitimate for the pure type-II model described in appendix D, since it does not rely in any kind of constraint. For this reason, we have moved this discussion to appendix D. To address the question of the number of excitations of the continuum model, we note that the two excitations are encoded in the presence of four fields $A_1,...,A_4$ entering the operator $e^{i \mathsf{T}KA}$ that creates excitations inside the physical subspace [Eq. (32) of the revised manuscript]. Upon a change of basis, these four fields can be transformed into two decoupled pairs of fields, with each pair corresponding to an independent degree of freedom. To highlight this point, we have introduced a new subsection, IIIC, in which we show that by introducing dynamical terms to the effective action, the two excitations are manifest trough the existence of two poles in the propagator of the theory. This is how the effective field theory captures the proper number of degrees of freedom of the lattice model.

---

## Round 2 · Referee Report · Anonymous (Referee 2) · 2021-7-3

Strengths

This paper proposed a continuum field theory description of a type-II fracton model, the Haah's cubic code. The novelty of the proposal is that the gauge field theory has a topological term similar to Chern-Simons theory, which differs from previous constructions.

Weaknesses

  1. The proposed derivation of the field theory from the lattice model starts from rewriting the Haah's code in terms of anti-commuting gamma matrices. Then a key step (Eq. (4)) is to represent the gamma matrices using a new variable A, which then becomes the dynamical field in the continuum limit. Given that this is the crucial passage from the lattice to field theory, it is unsatisfactory that the nature of this field A is never discussed. Naively A should be a discrete variable since gamma squares to 1, but from later discussions (especially from the discussion about excitations on page 10), I believe the authors treated A as a U(1) variable. This should be justified, or at least explicitly stated around Eq. (4).

  2. The introduction of the capital Gamma's is somehow not well motivated. Why is it necessary to have the Gamma's in place of gamma's to obtain the field theory? Or perhaps it reflects freedom in identifying the microscopic spin operators with the continuum field variables? It would be useful to have more explanations for this step.

== The following are not really "weaknesses", but rather questions/comments that I hope the authors can address ==

  1. Are the three conditions for the T vectors mutually exclusive, so they can never be satisfied simultaneously? This is the impression I got, and if so perhaps it should be explicitly stated.

  2. Is the choice for the T vector given in Eq. (10) unique? If not, how does the choice affect the resulting field theory?

  3. It is stated below Eq. (14) that the constraint leads to gauge structure, playing the role of Gauss's law in usual gauge theory. If this is the correct interpretation, then Eq. (14) should generate the gauge transformations given in Eq. (18), at least the time-independent ones. It would be good to clarify this point.

  4. A somewhat unusual feature of the field theory is that there are two "flavors" of the time component $A_0^{(\alpha)}$, $\alpha=1,2$, but not for the spatial components. I can understand the two flavors corresponding to two kinds of excitations (violations of the X term and the Z term) as $A_0$ is coupled to the charge density, but this interpretation would imply that there are two kinds of currents, so the spatial components should also have the flavor index. More discussions about this issue would help clarify the physical meaning of the gauge fields.

  5. The authors mentioned that the field theory is gapped. This is not entirely obvious from the continuum action given in Eq. (8). Perhaps an explicit calculation of the photon spectrum would be helpful.

  6. The constraint in Eq. (14) was required because in the lattice model, the eight operators on the corners of a cube multiply to identity. Since the constraint is crucial in establishing the immobility of excitations, one would wonder whether it is similarly significant in the lattice model.

Report

This work proposed an interesting solution to an important open problem in the theory of fracton topological order, namely continuum field theory for gapped type-II fracton model. A detailed derivation of the field theory was provided, and the authors showed that the field theory does have the right mobility structure for excitations. So I believe the paper meets the expectation for acceptance.

Requested changes

Please see above.

Minor issues:

  1. In the condition (ii) on page 5, "sistematically" should be "systematically".

  2. Below Eq. (5), the meaning of the index $\alpha$ should be explained.

  3. Near the end of the paragraph below Eq. (14), does the author mean "i.e. $c_{\vec{x}}^{(\alpha)}=-1\rightarrow q_{\vec{x}}^{(\alpha)}=1$"?

  • validity: high
  • significance: good
  • originality: high
  • clarity: good
  • formatting: excellent
  • grammar: -

Author:  Weslei Fontana  on 2021-10-19  [id 1866]

(in reply to Report 2 on 2021-07-03)

referee

The introduction of the capital Gamma's is somehow not well motivated. Why is it necessary to have the $\Gamma$'s in place of $\gamma$'s to obtain the field theory? Or perhaps it reflects the freedom in identifying the microscopic spin operators with the continuum field variables? It would be useful to have more explanations for this step.

response In the microscopic description, the $\Gamma$'s are completely equivalent to the $\gamma$'s, since they differ only by some combination of $\left(\gamma^I\right)^2=1$. On the other hand, in the continuum limit, working with $\Gamma$'s is necessary in order to reproduce the correct commutation relations (the $TKT$ conditions) of the cubic operators. Without the $\Gamma$'s, the $TKT$ conditions are not strictly zero and therefore break the gauge invariance of the continuum action. We included a paragraph in the main text to elucidate this point.

referee

Are the three conditions for the T vectors mutually exclusive, so they can never be satisfied simultaneously? This is the impression I got, and if so perhaps it should be explicitly stated.

response That is correct, i.e., they cannot be simultaneously satisfied. We included a comment in the main text about this point.

referee

Is the choice for the T vector given in Eq. (10) unique? If not, how does the choice affect the resulting field theory?

response The choices are not unique. The only criteria required is that they obey the $TKT$'s conditions such that they are all strictly zero. This ensures the gauge invariance of the continuum theory. Thus, there are equivalent sets of $T$-vectors, namely, any set of $T$-vectors with integer coefficients satisfying $TKT=0$ defines a consistent continuum theory.

referee

It is stated below Eq. (14) that the constraint leads to gauge structure, playing the role of Gauss's law in usual gauge theory. If this is the correct interpretation, then Eq. (14) should generate the gauge transformations given in Eq. (18), at least the time-independent ones. It would be good to clarify this point.

response The selection rule (17) [previously Eq. (14) in the original version] leads to a gauge structure in the sense that the cube operators become gauge invariant if we discard the first exponential in Eq. (14) of the revised paper. This is the reason why we have redefined the cube operators in Eq. (18); the original theory is recovered if condition (17) is satisfied. We made an effort to clarify this point in our revised version, in the paragraph below Eq. (17).

referee

A somewhat unusual feature of the field theory is that there are two "flavors" of the time component $A_0^{(\alpha)},\,\alpha=1,\,2.$, but not for the spatial components. I can understand the two flavors corresponding to two kinds of excitations (violations of the X term and the Z term) as $A_0$ is coupled to the charge density, but this interpretation would imply that there are two kinds of currents, so the spatial components should also have the flavor index. More discussions about this issue would help clarify the physical meaning of the gauge fields.

response The fact that the spatial currents have no flavor index means that the excitations in both cubes are not completely independent from each other. This is a feature present in the lattice model. For example, in the lattice we can act with a single $IY$ operator on a given site and this operator would produce two tetrahedral configurations (one in each cube). This feature is incorporated into the continuum theory through a spatial component of the current carrying no flavor index.

referee

The authors mentioned that the field theory is gapped. This is not entirely obvious from the continuum action given in Eq. (8). Perhaps an explicit calculation of the photon spectrum would be helpful.

response Thank you for this suggestion. By pursuing it, we have introduced a new section (Sec. IIIC) entirely dedicated to this point, which we think is very illuminating.

referee

The constraint in Eq. (14) was required because in the lattice model, the eight operators on the corners of a cube multiply to identity. Since the constraint is crucial in establishing the immobility of excitations, one would wonder whether it is similarly significant in the lattice model.

response The Haah code is an example of a CSS code. One main property of a CSS code is that the product of the operators on all corners of the code is the identity. In this sense, the fact that these operators multiply to the identity is of significant importance as part of the definition of the model itself. However, in the lattice model, this property does not necessarily ensure the immobility of the excitations. The immobility in the continuum theory is due to the requirement that all conditions (i), (ii), and (iii) are obeyed strictly, meaning that full immobility is a consequence of the collection of those conditions, not one of them in particular.

---

## Round 3 · Referee Report · Anonymous (Referee 1) · 2021-11-7

Report

I appreciate the changes the authors have made. I think it is ready for publication in SciPost.

---

## Round 3 · Author Response

First of all, we would like to apologize for the delay in providing this reply. We are very grateful to the referees for the pertinent criticisms. Following their comments, we have carefully revised the manuscript and addressed all the requested changes.

Sincerely,

The authors

---

## Editorial Decision

published